# PeerJ

# Too big to be noticed: cryptic invasion of Asian camel crickets in North American houses

Mary Jane Epps[1], Holly L. Menninger[1], Nathan LaSala[2] and Robert R. Dunn[1]

[1] Department of Biological Sciences, North Carolina State University, Raleigh, NC, USA
[2] Enloe High School, Raleigh, NC, USA

## ABSTRACT

Despite the rapid expansion of the built environment, we know little about the biology of species living in human-constructed habitats. Camel crickets (*Rhaphidophoridae*) are commonly observed in North American houses and include a range of native taxa as well as the Asian *Diestrammena asynamora* (Adelung), a species occasionally reported from houses though considered to be established only in greenhouses. We launched a continental-scale citizen science campaign to better understand the relative distributions and frequency of native and nonnative camel crickets in human homes across North America. Participants contributed survey data about the presence or absence of camel crickets in homes, as well as photographs and specimens of camel crickets allowing us to identify the major genera and/or species in and around houses. Together, these data offer insight into the geographical distribution of camel crickets as a presence in homes, as well as the relative frequency and distribution of native and nonnative camel crickets encountered in houses. In so doing, we show that the exotic *Diestrammena asynamora* not only has become a common presence in eastern houses, but is found in these environments far more frequently than native camel crickets. Supplemental pitfall trapping along transects in 10 urban yards in Raleigh, NC revealed that *D. asynamora* can be extremely abundant locally around some homes, with as many as 52 individuals collected from pitfalls in a single yard over two days of sampling. The number of *D. asynamora* individuals present in a trap was negatively correlated with the trap's distance from a house, suggesting that these insects may be preferentially associated with houses but also are present outside. In addition, we report the establishment in the northeastern United States of a second exotic species, putatively *Diestrammena japanica* Blatchley, which was previously undocumented in the literature. Our results offer new insight into the relative frequency and distribution of camel crickets living in human homes, and emphasize the importance of the built environment as habitat for two little-known invading species of Orthoptera.

Corresponding author
Mary Jane Epps, mjepps@ncsu.edu

## INTRODUCTION

In the United States, 90% of the human population is predicted to live in urban environments by 2050 (*United Nations, 2012*). At this time a large geographic area will be urban, peri-urban and suburban (*Nowak & Walton, 2005*), an area greater than that covered by many of North America's primary vegetation types (*Stein, Kutner & Adams, 2000*). Although the species living in built environments are among the organisms we see most often, they are not necessarily well-documented. We suspect this is particularly true of species that are neither valued aesthetically, as are birds and butterflies, nor are important economic pests (such as bed bugs and roaches). Camel crickets (Orthoptera: *Rhaphidophoridae*) are among the largest of the many insects that live in modern-day houses, and have an especially longstanding history of contact with humans in our homes. These insects have long been noted in basements and cellars; with one remarkable example of cave art from Paleolithic France depicting the cave-dwelling camel cricket *Trogophilus* sp. (*Chopard, 1928*). The relationship between camel crickets and humans is clearly ancient; however, the biology of these insects as residents of our homes is known primarily from a smattering of specimen records in museums rather than from formal study.

Camel crickets comprise a moderately diverse family of Orthoptera, represented by ca. 150 species (23 genera) across North America (*Arnett, 2000*). Of these, several species in the large genus *Ceuthophilus* Scudder (e.g., *C. brevipes* Scudder, *C. pallescens* Bruner, *C. agassizii* (Scudder), *C. latens* Scudder, *C. maculatus* (Harris), and others) have been reported as common or occasional inhabitants of North American homes, particularly in cellars and basements (*Blatchley, 1920*; *Vickery & Kevan, 1983*). Outside of these habitats, most species of *Ceuthophilus* (including those occurring in our houses) are found under rocks, logs, or surface debris in forested areas, although a few are known from grassland ecosystems (*Vickery & Kevan, 1983*). Other *Ceuthophilus* (e.g., *C. carlsbadensis* Caudell, *C. longipes* Caudell, *C. secretus* Scudder, and others) are common residents of caves, where the droppings and carcasses of these crickets are a major source of energy for other organisms. For this reason camel crickets are considered keystone species in many cave ecosystems (*Lavoie, Helf & Poulson, 2007*; *Taylor, Krejca & Denight, 2005*). Although cellars and basements share features with caves (e.g., both tend to be damp, dark, and low in nutrients), most camel crickets found in our houses appear to be distinct from species typically collected in caves (*Vickery & Kevan, 1983*). However, in at least one case (a camel cricket endemic to Tuscany, Italy, *Dolichopoda schiavazzii* Capra), populations of an otherwise cave-inhabiting species are also known from cellars and other subterranean environments of human origin (*Allegrucci, Minasi & Sbordoni, 1997*).

In addition to a rich diversity of native camel cricket species, nonnative camel crickets have also become established in North America. The 'greenhouse camel cricket,' *Diestrammena asynamora* (Adelung) is a species native either to Japan or the Sichuan region of China (*Rehn, 1944*). This species was first recorded in North America in 1898 from a greenhouse in Minnesota (*Rehn, 1944*), and subsequently has been noted in a number of locations across the eastern and central United States and Canada. *D. asynamora* has also been found throughout much of Europe (*Rehn, 1944*). Many authors have considered this

species to be associated primarily with greenhouses (e.g., *Bue & Munro, 1939*; *Rehn, 1944*; *Vickery & Kevan, 1983*), although a few early reports document at least an occasional presence in cellars (*Blatchley, 1920*; *Rehn, 1944*). However, little discussion has been made of this species and its status since *Rehn*'s *1944* publication. Modern reports of *D. asynamora* show that it is present in some basements, though it is uncertain whether the sightings of this introduced cricket represent isolated cases of localized abundance or a more extensive invasion. Recent anecdotal reports (www.bugguide.net) also suggest the establishment of a second Asian species, *D. japanica* Blatchley (syn. *D. japonica* Karny, *D. naganoensis* Mori), around New York City, NY. Because camel crickets include both introduced species and geographically and locally rare species, it is possible that basements and cellars might be important habitats for the spread of introduced camel crickets and/or the persistence of native camel cricket species.

One challenge with studying the biology of species living in homes is that privacy concerns make these areas difficult to sample. However, citizen science may offer an ideal approach for studying home biodiversity; volunteers can participate in scientific research by self-surveying their own homes. Although obtaining accurate identifications of organisms from public survey data can be challenging, many of the characters that distinguish camel crickets at the generic and/or species level (e.g., coloration, tibial armature) may be visible in photographs. For this reason, photographic documentation is an invaluable addition to public survey data, and provides an easy way to confirm the presence and distribution of camel crickets in our homes.

In this study, we use citizen-contributed data to offer new insight into the distribution and composition of camel crickets taking shelter in human homes. Initially, in order to understand how common camel crickets are in houses, we surveyed citizens across the United States about the presence of camel crickets living in and around their homes. We conducted this survey in two ways: (1) we asked visitors to our website to report the presence/absence of camel crickets (as well as other natural history observations in and around their home) via an open survey and (2) we directly administered a closed survey that included a question about camel crickets to volunteers wishing to participate in an unrelated citizen science project (about microbial diversity in the home). We then solicited photographs and specimens of camel crickets from citizen scientists to evaluate the occurrence and geographical distribution of native versus nonnative camel cricket species in homes. These survey results were augmented with trapping efforts to compare the composition of camel crickets living in houses to those present in urban yards.

## METHODS

We used two types of citizen science surveys to characterize the geographic distribution and composition of camel crickets in houses across the United States. First, as part of a broader study about the ecology of human homes, we used an open web survey to poll people across the United States about the organisms they find in their homes; this survey included questions about the presence/absence of camel crickets in or around their homes and their geographic location (Appendix A). We recruited participants to the survey

through our website (yourwildlife.org), social media and email campaigns, and the survey remained open to public responses from December 2011 through July 2013. This initial survey had the potential to be biased toward individuals who had camel crickets in their homes, as people may be more likely to report a presence than absence (*Bonney et al., 2009*). As a result, we conducted a second survey by polling a geographically stratified but naïve population of homeowners. We directly administered a closed survey to 7,058 households wishing to enroll in the Wild Life of Our Homes project (WLOH, a separate study mapping the indoor microbial biodiversity of homes; homes.yourwildlife.org) over the period October 2012–April 2013. Volunteers, representing all 50 states and the District of Columbia, were required to complete the brief survey (containing a question about the presence/absence of camel crickets in the home, Appendix A) in order to receive a home microbe sampling kit. Thus, participation was not a function of initial interest in camel crickets, but in this other citizen science project, therefore reducing sampling bias.

The results of both surveys were used to map the presence of camel crickets in North American homes. Maps were created using ArcGIS software (*ESRI, 2006*) and R (www.r-project.org/). Only data from the second, WLOH participant survey were used to estimate the prevalence of the crickets.

In order to understand the relative distribution of native versus nonnative camel crickets we next solicited photographs and/or specimens of camel crickets from citizen volunteers who reported these insects in their homes. These volunteers included a subset of participants from the surveys described above, as well as additional individuals responding to an appeal for participation on the Camel Cricket Census website (crickets.yourwildlife.org/). Photographs and specimens were identified to genus based on tibial armature and other relevant characters described in *Vickery & Kevan (1983)*. Where possible, photographs and specimens of the nonnative *Diestrammena* were further identified to species using characters such as the number of tibial spines, tibial spur length, and color pattern as described in *Sugimoto & Ichikawa (2003)*, *Vickery & Kevan (1983)*, and following consultation with experts.

Finally, to understand whether the Asian camel cricket *D. asynamora* is living only in houses or also present in yard habitats, we sampled camel crickets in a subset of urban yards at increasing distances from homes known to contain camel crickets. In July of 2013, 10 participating households were recruited from central Raleigh, NC, and pitfall traps placed in the yard of each. Study yards were located within a 1.5 mile radius of North Carolina State University's central campus, and were typically a mix of sun and shade habitat with occasional scattered trees and primarily grass as groundcover. We constructed pitfall traps using plastic cups (7 cm across by 10 cm deep) and placed three traps per yard at distances of 1 m, 4 m, and 8 m from each house along a haphazardly placed transect. Traps were baited with a 1:1 dilution of molasses and water as per the methods of *Hubbell (1936)*. We placed inverted plastic bowls elevated approximately 3 cm over the mouth of each trap to protect traps from rain and to encourage camel cricket visitation by offering cover. Contents of traps were collected daily and traps were left in place for two days. In some yards small mammals would disturb the traps, in which case we replaced the

molasses bait with soapy water in all traps on a transect to be less attractive to mammalian pests. We sorted the contents of each trap in the laboratory and identified all camel crickets to species with the aid of a dissecting microscope. We then used analysis of covariance (ANCOVA) to test for a relationship between the number of *D. asynamora* individuals in a trap and its distance from a house. Yards in which no camel crickets were recovered at any of the three traps were excluded from analysis. Statistical analysis was performed in JMP v. 10.0 (SAS Institute, Cary, NC).

## RESULTS

Individuals from 549 homes responded to our initial open survey question about the presence or absence of camel crickets in or around their houses, offering positive reports of camel crickets for 244 homes across the country. An additional 1,719 households responded to the unbiased (with respect to camel cricket presence) WLOH participant questionnaire. Over both surveys, participants from 669 houses reported having observed camel crickets in their homes, including 24.4% of households responding to the unbiased WLOH study (Fig. 1). Together, these surveys allowed us to evaluate the potential distribution of camel crickets associated with human houses across the United States. We use the word "detection" to acknowledge that reports of absence in survey data may reflect failures to detect camel crickets, just as presences may represent failures at identification. For example, a large spider might bear a vague resemblance to a camel cricket for a participant wary of arthropods. Participants from 39 states and the District of Columbia reported observing camel crickets in or around their homes (Fig. 2). The proportion of detections of camel crickets in homes was significantly higher in the eastern United States (28% of reports were positive from states east of Colorado) compared to western states (7% positive reports; two-tailed $P < 0.0001$ from Fisher's exact test). Based on the proportion of photographs showing insects incorrectly identified as camel crickets by citizen scientists who responded to our call for photographs (see below), we estimated a 4.6% error rate associated with affirmative reports of camel crickets from all survey data, although this error rate may vary geographically. From Colorado westward only five photographs were submitted, of which 40% (2 of 5) were identified incorrectly as *Rhaphidophoridae*. The most common taxa mistaken for camel crickets were field crickets (Gryllidae) or other Orthoptera.

Citizen scientists from 163 households submitted identifiable photographs ($N = 151$) and/or specimens ($N = 12$) of camel crickets from their houses. Submissions spanned 23 states and the District of Columbia, as well as one Canadian province; with an overrepresentation of submissions (37%) from North Carolina. Out of all identifiable camel cricket submissions, 88% of houses submitted evidence of the Asian genus *Diestrammena*. Only 12% of houses reported members of the native *Ceuthophilus* (Table 1 and Fig. 3). In three cases, evidence of both native and nonnative genera were contributed from the same home. Of the 143 submissions recognizable as *Diestrammena*, 108 were of sufficient quality to allow identification to species. Of these, 94% of entries were *D. asynamora* (Fig. 4A), while the remainder (seven entries) were identified as putative *D. japanica* (Fig. 4B). Records of

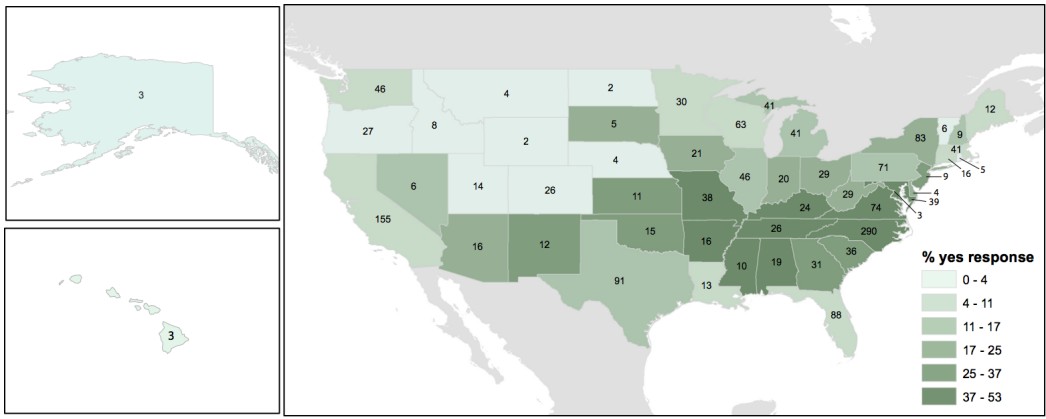

**Figure 1 Percentage of households by state reporting the presence of camel crickets around the home.** Responses to the Wild Life of Our Home survey, showing the percentage of households from each state answering 'yes' to the question 'have you seen camel crickets in or around your home?'. Numerals in each state represent the total number of responses (yes or no) for that state.

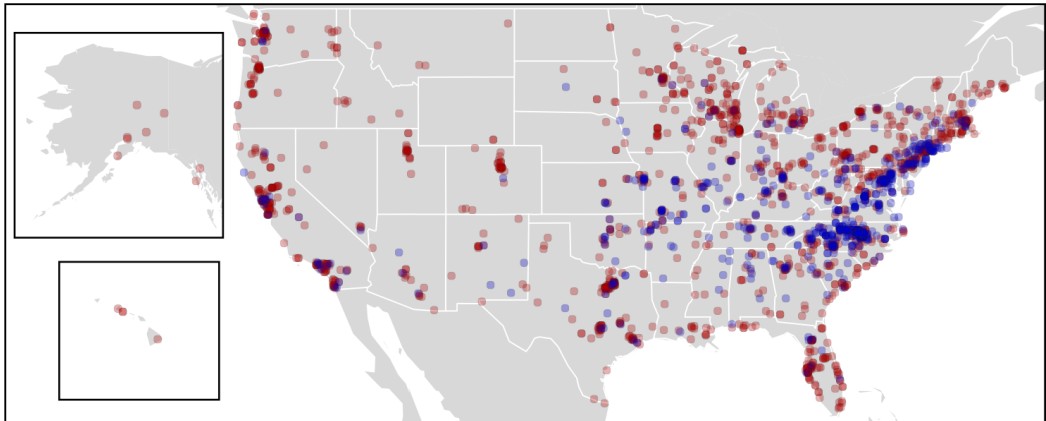

**Figure 2 Survey responses showing presence or absence of camel crickets in homes.** Map of combined responses to the open survey question and the Wild Life of Our Homes survey question asking citizens if they have observed camel crickets in their houses. Blue points represent positive reports of camel crickets found in homes ($N = 669$), whereas red points indicate households where camel crickets have not been knowingly observed ($N = 1,598$).

*D. japonica* were submitted exclusively from the northeastern United States in Massachusetts, Pennsylvania, and New Jersey (Fig. 5).

We recovered 158 camel crickets from pitfall traps in urban yards in Raleigh, NC. Prior to our investigation, eight of the 10 households participating in our trapping study reported previously having seen camel crickets in their home. Camel crickets were found in seven of the yards sampled, and were absent in both of the yards for which camel crickets were not reported in the house. For houses initially reporting camel crickets as present, an average of 20 individual camel crickets were recovered per yard over the two-day sampling period (95% CI [5–34], range = 0–52). All recovered specimens of *Rhaphidophoridae* were identified as the Asian species *D. asynamora*. The number of

Table 1 **Number of houses with camel crickets by state.** Results of citizen-contributed photographic or specimen submissions showing the relative number of households with the Asian *Diestrammena* versus native *Ceuthophilus* samples by state or Canadian province. Two houses in North Carolina each contributed specimens of both genera. The subset of households submitting photographs and/or specimens of *Diestrammena* that could be determined to species are further distinguished to show the relative number and distribution of records for *D. asynamora* versus putative *D. japanica*.

| State or province | Total responses | # Houses with *Ceuthophilus* | # Houses with *D. asynamora* | # Houses with *D. japanica* | # Houses with *Diestrammena* sp. (unidentified) |
|---|---|---|---|---|---|
| Colorado | 2 | 2 | 0 | 0 | 0 |
| Delaware | 2 | 0 | 1 | 0 | 1 |
| District of Columbia | 1 | 0 | 0 | 0 | 1 |
| Georgia | 2 | 0 | 2 | 0 | 0 |
| Iowa | 1 | 1 | 0 | 0 | 0 |
| Illinois | 2 | 0 | 2 | 0 | 0 |
| Kansas | 2 | 1 | 1 | 0 | 0 |
| Kentucky | 1 | 0 | 0 | 0 | 1 |
| Massachusetts | 3 | 1 | 0 | 2 | 0 |
| Maryland | 17 | 1 | 10 | 0 | 6 |
| Michigan | 1 | 0 | 0 | 0 | 1 |
| Missouri | 9 | 4 | 5 | 0 | 0 |
| North Carolina | 60 | 5 | 41 | 0 | 14 |
| New Hampshire | 1 | 1 | 0 | 0 | 0 |
| New Jersey | 16 | 0 | 12 | 2 | 2 |
| New York | 9 | 0 | 7 | 0 | 2 |
| Ohio | 2 | 0 | 2 | 0 | 0 |
| Pennsylvania | 10 | 2 | 4 | 2 | 2 |
| Saskatchewan | 1 | 1 | 0 | 0 | 0 |
| South Carolina | 1 | 0 | 1 | 0 | 0 |
| Tennessee | 3 | 0 | 3 | 0 | 0 |
| Texas | 1 | 1 | 0 | 0 | 0 |
| Virginia | 15 | 0 | 10 | 0 | 5 |
| Wisconsin | 1 | 0 | 1 | 0 | 0 |
| Total records | 163 | 20 | 102 | 6 | 35 |

*D. asynamora* individuals recovered in traps was negatively correlated with a trap's distance from a house ($R^2 = 0.66$, $P = 0.004$ from ANCOVA after ln-transformation of the number of cricket individuals and after accounting for individual yard, as would reflect variation in local abundances of camel crickets; Fig. 6). In fact, for every yard with successful trapping, more crickets were consistently recovered at traps 1 m from the house than were recovered from the two more distant traps combined. However, in 57% of these yards at least one *D. asynamora* individual was recovered from the trap placed farthest from the house (8 m).

## DISCUSSION

Although camel crickets have long been a common presence in our homes, little is known about the identity, occupancy and geography of these animals in homes. Using data

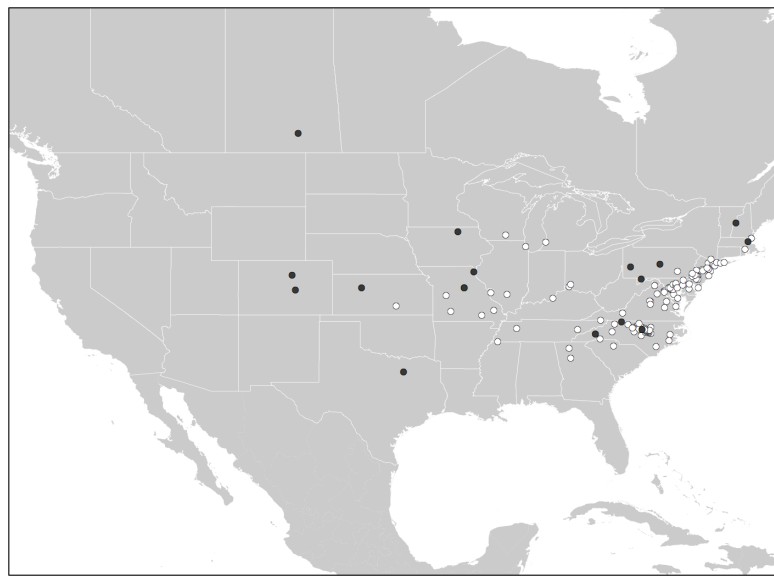

**Figure 3 Distribution of *Ceuthophilus* and *Diestrammena* in homes.** The distribution of native *Ceuthophilus* spp. (black circles; $N = 20$) versus exotic *Diestrammena* spp. camel crickets (white points; $N = 143$) in homes, based on photographic and specimen submissions contributed by citizen scientists.

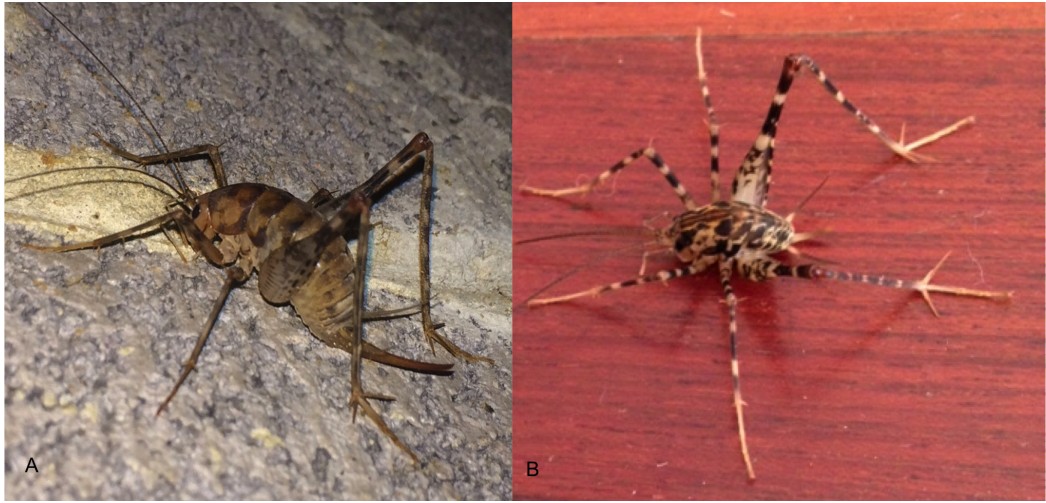

**Figure 4 Images of *Diestrammena asynamora* and putative *D. japanica*.** Photographs of the two species of *Diestrammena* submitted by citizen scientists, showing (A) *D. asynamora* (Andrew Blanchard, Creative Commons Attribution License 2014) and (B) putative *D. japanica* (Kathryn Kinney, Creative Commons Attribution License 2014).

contributed by citizen scientists, we found that camel crickets are common in houses across much of the continental United States (Fig. 2), and present in as many as a quarter of homes surveyed as part of another citizen science project. Based on our survey results, camel crickets appear to be geographically widespread in homes particularly across the eastern half of the United States, and to a lesser extent in the southwest and west coast (Fig. 1). Camel crickets were not reported in homes throughout much of the mountain

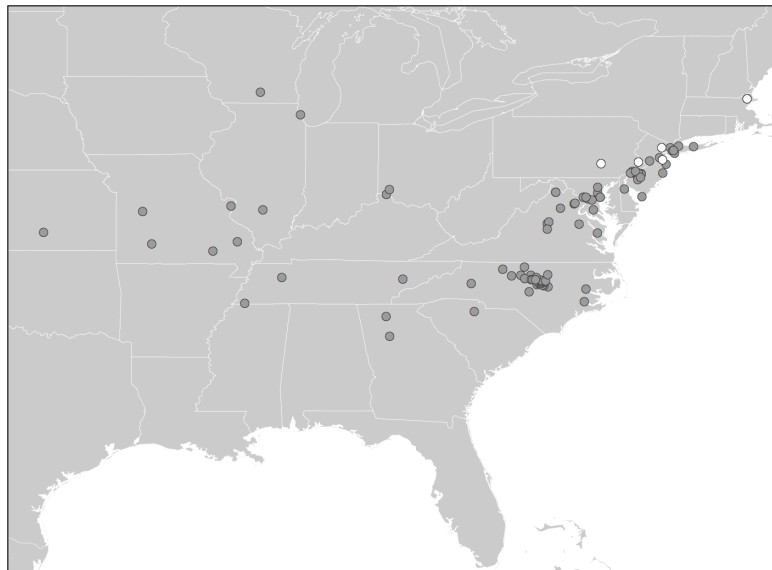

**Figure 5 Distribution of *Diestrammena* species in American houses.** The relative distribution of the two *Diestrammena* species reported from houses in the United States, as indicated by photographs and/or specimens from citizen scientists. Records of *D. asynamora* ($N = 101$) are indicated in grey, *D. japanica* ($N = 7$) in white.

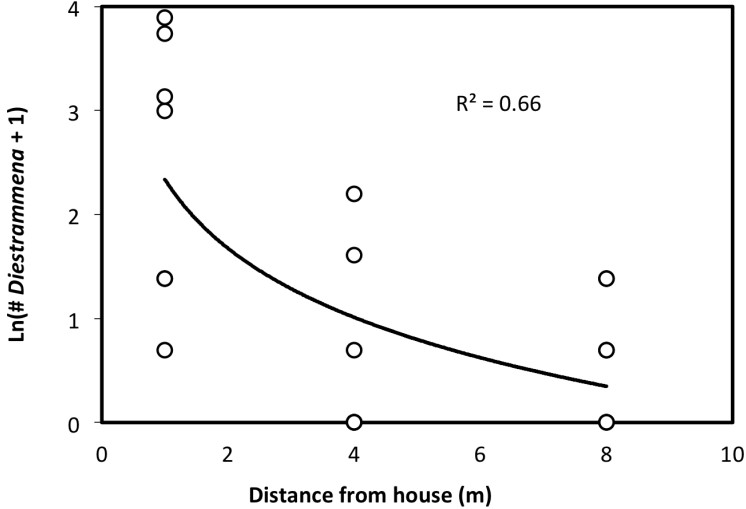

**Figure 6 The number of *D. asynamora* crickets in yard traps as a function of the distance from a house.** The number of *D. asynamora* individuals recovered from a pitfall trap was negatively correlated with the distance of a trap from a house ($R^2 = 0.66$, $P = 0.004$ from ANCOVA, after accounting for individual yard effect). Each dot represents a single pitfall trap ($N = 21$, with three traps placed in each of seven yards).

west, although responses from this region were sparse relative to more populated parts of the country (Figs. 1 and 2). Hence, there may be large areas in the west where camel crickets are more common than our data indicate. However, the public can still contribute data to this project, such that we hope to add data for the western US in the coming months or years (crickets.yourwildlife.org). Camel crickets appear to be most prevalent in houses in the southeastern United States, with nearly 50% of households surveyed in Virginia, North Carolina, Missouri, Mississippi, Maryland, and Tennessee reporting the presence of camel crickets in their homes (Fig. 1).

Although this survey was potentially biased geographically, it was not biased as a function of the presence or absence of camel crickets. If the abundance of camel crickets (of any species) in and around the average home where they are present is comparable to the abundance of these insects recovered in our pitfall traps around homes in Raleigh, North Carolina, there could be as many as seven hundred million camel crickets in and around homes across the eastern United States alone (calculated based on the number of housing units reported in the 2011 US Census for midwest, northeast and south US census regions for which our survey responses were generally high; see Fig. 2; *US Census Bureau, 2012*). If correct, this would amount to more camel crickets than humans. Although clearly a very rough estimate (e.g., we do not account for other factors such as regional geographic variation in abundance), this figure nonetheless offers a rough estimate of the large populations of camel crickets that may have become established in and around built environments. The size of these populations is all the more remarkable when we consider that most of these camel crickets belong to an introduced species previously not known to be especially common; in contrast, native species appear to be comparatively rare in these environments.

Citizen scientists' submissions of photographs and specimens of camel crickets found in homes reveal that the Asian camel cricket *D. asynamora* has become a successful and widespread invader throughout the eastern United States (Fig. 3). Across much of this region this species appears to be a much more common occupant of human homes compared to native *Ceuthophilus* spp. (Table 1). For example, in North Carolina, the state for which we have the richest data, *Diestrammena* (representing *D. asynamora* in all identifiable entries) was present in 92% of houses with camel cricket samples submitted (Table 1). Our pitfall trapping in urban yards reveals that this species also can be extremely abundant, with more than 50 individuals found over two days of sampling in a single yard in Raleigh, North Carolina.

Although *D. asynamora* is clearly widespread and abundant in the eastern United States, the extent of this species' range outside of the eastern United States is unclear. *Rehn (1944)* describes reports of the species in greenhouses and cellars from Maine south to Tennessee, and as far west as Colorado. While his reports derive from a small number of museum specimens combined with scattered anecdotes, they already cover a relatively large geographic area. Our study includes reports of the species only as far west as Kansas, though extending farther south into Georgia and South Carolina. Other reports suggest an even larger distribution, but lack of specimen data makes comparison to our results

**Peer**J

difficult (*Vickery & Kevan, 1983*). According to most accounts, established populations of this species were thought to be present in greenhouses only (*Blatchley, 1920*; *Vickery & Kevan, 1967*; *Vickery & Kevan, 1983*). Hence, *D. asynamora* likely may have increased in abundance since 1944, particularly in houses; however, the species does not necessarily appear to have expanded its geographic distribution.

Despite the abundance of *D. asynamora* in and around the home, little is known about the cricket's habits and life history. This species is best known in the literature from its occurrence in greenhouses, where it has been blamed for causing occasional minor damage to plants (*Vickery & Kevan, 1983*). Although its feeding preferences are largely unknown, *D. asynamora* appears like most *Rhaphidophoridae* to be an omnivorous scavenger, and has been reported foraging on living and dead plant matter and dead insects. The habits and dietary preferences of our native *Ceuthophilus* spp. are only somewhat better known than those of *D. asynamora*. Species of *Ceuthophilus* (including taxa sometimes found in houses) have been observed scavenging opportunistically on a range of food sources including other insects, fungi, and fallen fruit (*Taylor, Krejca & Denight, 2005*). Examination of several cave-associated *Ceuthophilus* species (*Northup, 1988*) revealed a diverse diet that included mammalian carcasses, feces, other insects (including cannibalized *Ceuthophilus* individuals), and human food waste. Like *D. asynamora*, some species of *Ceuthophilus* appear to be occasional predators of other insects (*Taylor, Krejca & Denight, 2005*; *Vickery & Kevan, 1983*). Although we know little about the life history of *D. asynamora* or *Ceuthophilus* in houses, *D. asynamora* is reported to breed year-round in heated greenhouses. Breeding in *D. asynamora* is thought to occur only in the dark, and eggs are typically laid in the soil (*Vickery & Kevan, 1983*).

Although *Ceuthophilus* and *Diestrammena* both appear to be fairly generalized omnivores, we can infer little about the extent to which these insects' habits and life histories are comparable (and hence might lead to direct or indirect competition) in houses. Differences in the natural history of the two taxa potentially could affect the extent to which these crickets would be perceived by our citizen scientist contributors. For example, if the principal species of *Ceuthophilus* in houses were found to be more reclusive than *D. asynamora*, this could in part explain the larger number of *D. asynamora* reports by our contributors. However, the results of our pitfall experiment suggest that for at least some areas the perception of greater numbers of *D. asynamora* than *Ceuthophilus* associated with houses is real, especially considering that numerous individuals of various *Ceuthophilus* species (including those sometimes found in houses) have been commonly recovered from molasses-baited pitfall traps in other studies not near houses (*Blatchley, 1920*; *Hubbell, 1936*; *Taylor, Krejca & Denight, 2005*). Although we might expect larger insects to be noticed more readily by citizen scientists, the principal species of *Ceuthophilus* occurring in houses are in fact slightly larger on average than *D. asynamora* (*Blatchley, 1920*). Hence, the more numerous reports of *D. asynamora* relative to *Ceuthophilus* do not appear to have been biased by body size. Nonetheless, we recommend future study to understand the relative life histories of these species, their interactions with each other and other house-dwelling arthropods, and the ways in which they use our houses as habitat.

The appearance of a second introduced species, *Diestrammena japanica,* as an exotic in the United States has never before been recorded in the literature, although its presence has been reported anecdotally in some northeastern states concordant with those found in our study (see www.bugguide.net). Some confusion arises in the erroneous early use of the name *D. japanica* as a synonym for *D. asynamora* (*Blatchley, 1920*), a misapplication subsequently clarified by *Rehn (1944)*. Aside from *D. asynamora* only one other species of *Diestrammena* has been reported in the literature from the United States. This second species, reported by *Morse (1904)* as *D. unicolor*, is known in the United States from only a single specimen collected in a greenhouse in Chicago. This specimen, described as being uniformly piceous in color (*Blatchley, 1920*), is clearly distinct from our records of *D. japanica*, despite any potential nomenclatural incongruities.

In our study, the presence of two species of *Diestrammena* in our samples is confirmed by the widely divergent number of tibial spines between *D. asynamora* and *D. japanica* (ca. 60 and 30 respectively; *Sugimoto & Ichikawa, 2003*), a character clearly visible in many of the photographs submitted. Although our identification of the species *D. asynamora* was confirmed on the basis of at least 170 specimens as well as by photographs, our records for *D. japanica* were based on photographs only, and we must therefore allow for some uncertainty as to the identity of the second species as the true *D. japanica*. Photographs of the second *Diestrammena* species were identified as *D. japanica* based on a combination of outwardly visible characters such as tibial spur length, tibial spine number, pronotal luster, and coloration (*Sugimoto & Ichikawa, 2003*). In addition, the distinctive pronotal pattern of *D. japanica* (*Sugimoto & Ichikawa, 2003*) was an exact match for our specimens. Although we offer clear evidence for the presence of two introduced species of *Diestrammena* in the United States, we recommend future study of museum specimens and examination of male genitalia (ideally in comparison with type material) to confirm the second species as *D. japanica*.

Our collections of camel crickets from pitfall traps in urban yards revealed that *D. asynamora* is not restricted to house environments, but is also a common presence in adjacent yards. Whether these same individuals present in yards are also moving in and out of houses is unclear. The fact that these crickets were significantly more abundant in traps placed within a meter of the house (Fig. 6) suggests that *D. asynamora* may be closely associated with human dwellings even when found in outdoor habitats. However, as we did not account for potential variability in local habitats within a transect (e.g., traps placed 1m from houses could have been more proximate to bushes compared to further traps), such variation may have biased our transect results. Surprisingly, no native camel crickets were recovered from any of these traps. Although behavioral differences (e.g., in activity level) between cricket species could potentially cause one species to appear more readily in pitfalls, pitfall traps baited with molasses have repeatedly been shown in other work to be highly profitable for sampling *Ceuthophilus* spp. (*Blatchley, 1920*; *Hubbell, 1936*; *Taylor, Krejca & Denight, 2005*). This suggests that in some localities *D. asynamora* may be the dominant camel cricket not only in houses but also in urban yards. However, it is yet unclear whether *D. asynamora* has also invaded wilder habitats with less human

disturbance, or if in North America the species persists exclusively in habitats associated with anthropogenic structures. The extent to which *D. asynamora* has actually displaced or is actively competing with native populations of *Ceuthophilus* (a genus that includes a number of rare or sensitive species) is also unknown, and further study is needed to determine whether this new invader poses an ecological threat, or is merely a harmless visitor in our houses and yards.

## ACKNOWLEDGEMENTS

We would like to thank Lauren Nichols for mapping assistance, and David Ferguson and Piotr Naskrecki for initial advice on *Diestrammena* identification. Piotr Naskrecki, Terry Wheeler, and two anonymous reviewers provided helpful comments on the manuscript. We are also grateful to Lea Shell for help with public outreach and data management, and to Neil McCoy for technical and design assistance with the project website and associated data collection. Andrew Blanchard and Kathy Kinney kindly allowed us use of their photographs for Figs. 4A and 4B, respectively. We extend our warmest thanks to the many citizen scientists and survey participants who aided this project, without whose help this study would not have been possible. We also thank Bill Reynolds from the NC Museum of Natural Sciences for assistance with project outreach.

### Funding

This work was funded by the National Science Foundation (NSF 0953390 to RRD). The funders had no role in study design, data collection and analysis, decision to publish, or preparation of the manuscript.

### Grant Disclosures

The following grant information was disclosed by the authors:
NSF: 0953390.

### Competing Interests

The authors declare there are no competing interests.

### Author Contributions

- Mary Jane Epps conceived and designed the experiments, performed the experiments, analyzed the data, wrote the paper, prepared figures and/or tables.
- Holly L. Menninger conceived and designed the experiments, performed the experiments, reviewed drafts of the paper.
- Nathan LaSala performed the experiments.
- Robert R. Dunn conceived and designed the experiments, contributed reagents/materials/analysis tools, wrote the paper, prepared figures and/or tables, reviewed drafts of the paper.

## Supplemental Information

Supplemental information for this article can be found online at http://dx.doi.org/10.7717/peerj.523#supplemental-information.

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
