# Peer review of "Too big to be noticed: cryptic invasion of Asian camel crickets in North American houses"

_PeerJ, doi:10.7717/peerj.523_

## Round 0.1 · original submission · Major Revisions

First of all, I'd like to thank all three referees for their insightful, professional, and expert assessment of this MS. All three reviewers are excellent entomologists and natural historians, and all three work directly and/or indirectly with citizen science initiatives. To be honest, these three constitute one of the best trios of experts that could have been assembled to review this MS. I am sincerely grateful for their contributions.

For sake of history and full disclosure, I first asked Reviewer 1 and Reviewer 2 for reviews. I received a split decision: minor revisions vs. rejection. Both referees made excellent points in their respective reviews, and the authors need to carefully consider all of those points in their revisions.

Following the split decision, I contacted Reviewer 3 and asked for a third review. I provided Reviewer 3 with the anonymous text (cut-and-pasted from a TXT file) of the two previous reviews so that they could review the paper in the context of the reviews.

Reviewer 3 returned a decision of minor revisions as well, adding a variety of excellent suggestions to improve the MS. Again, the authors need to be sure to address these concerns, particularly as some of them are similar to the critiques offered by Reviewer 2 and Reviewer 1.

I also read the entire paper and can confidently say that all of the reviewers have made excellent points. While the trio of reviews tilts toward minor revisions, I believe that the number and depth of revisions (or rebuttals) that are required tends to push the decision into major revisions territory.

I would strongly encourage the authors to also make the review history of this paper public. Doing such would be in the spirit of citizen science; would allow others to consider the valid criticism of the study; and would serve as a teaching tool for how peer-reviewed science is conducted, with good and constructive criticism being used to improve a MS.

Thank you to the authors both for submitting this MS to PeerJ and for conducting the interesting – yet no doubt at times difficult – work of running a citizen science initiative. In an era of increasing tethers to screen-based technology, initiatives like this are vital to encourage the appreciation of natural history. It is only by fostering such interest among all ages that we will truly see changes in attitudes and behaviors toward the genuine care for the earth an all its diverse denizens.

Once again, thank you to all three reviewers. I truly appreciate the effort that you made in assessing this MS and look forward to working with each of you again in the future.

Reviewer 1 ·

Basic reporting

Nice article about a citizen science survey of Camel Crickets in North America, with data validation and a comprehensive background. The language is clear and concise. Known camel cricket distributions are described thoroughly, but no mention is made of their basic life history. For example, what do they eat? Why are they common in caves and basements?

Experimental design

Experimental design is original and creative, and clearly reported. Surveys that rely on homeowners’ self-reporting of camel cricket presence are likely to be biased, but this is noted in the MS. Further, the authors make an effort to quantify error rate based on photos that accompany participants’ identifications.
One point that is not mentioned is the size and behavior of the different camel cricket species. If the introduced species is much bigger or less retiring than the native species, then this would be reflected in the data as the non-native species being much more common. The same is true for activity levels. If one species is more active, they will be more commonly collected in pitfall traps. These points should be acknowledged and taken into account when interpreting the data.

Validity of the findings

The findings are valid, previously unreported and interesting. There is some speculation as to the geographic range and total number of individuals of Camel Crickets, but this speculation is clearly identified as a rough calculation based on the data collected. The authors should take into account some behavioral aspects of the different species (noted above) that could bias relative numbers detected, photographed and collected.

Additional comments

Excellent use of the citizen science approach. Wonderful to see that one of the coauthors is a high school student!

Reviewer 2 ·

Basic reporting

This paper explores the interesting topics of urban ecology and citizen science. In particular, the distribution of camel crickets in North American homes is investigated primarily using data generated through public web surveys and contributed photographs. What is established is that most records of camel crickets come from eastern North America. Also some information is derived about the taxa of camel crickets that are observed. In addition to the citzen-scientist-generated data, transect surveys of 10 yards in Raleigh, NC are used to measure the abundance of these insects near to and far away from homes. Unfortunately, the amount and quality of information submitted does not warrant publication at this time. Really, we are left with some distribution maps of camel crickets, some photographic identifications of particular species, and an inference that there are more crickets near houses than farther away. There are possible problems in several areas including (1) the quality of citizen scientist reporting, (2) an eastern bias to respondents to to surveys, and (3) attempts to identify species completely via photograph. A few suggestions for improvement: (1) It may make more sense to report more detailed information from the area where the survey responses are consistently high vs. attempting to make continent-wide assertions re. distributions, (2) Species-level identifications via photograph need to be backed up with specimen-based identifications, (3) If identifications are considered accurate, it may be possible to compare cricket communities among different areas, (4) The transect work is a good start, but could be expanded to more sites. Mark-recapture experiments might also give insight into the patterns of movement of the crickets, (5) The paper is presented completely without any information on the natural history or life history of camel crickets. What do these insects eat? Why do they inhabit homes? Does their eastern distribution in homes reflect their natural distribution? There is really very little about camel cricket biology in the paper.

Experimental design

Although I am a proponent of citizen science, I am somewhat concerned about unconfirmed identifications of insects by the public. Some follow up and/or training of collectors seems warranted. Also, the uneven distribution of responses to the survey may have affected the results presented. As I indicated above, making identifications from photographs should be supplemented by collection of specimens in the areas where the photos originate. This would validate the photo identifications. Finally, the transect study is interesting, but, as this is the only "experimental" or quantitative data in the paper, this study could have been expanded.

Validity of the findings

The findings of the study are valid if assumptions are made about the accuracy of citizen scientist reporting. I would have had more confidence in the results if training, follow up and validation of public identifications were done. Photo identifications need confirmation with specimens as I've already indicated. The transect data are interesting but are from only 10 sites. No information is derived that addresses whether the insects enter or leave homes for foraging.

Additional comments

Overall, I believe that, although urban entomology and citizen science are valid areas of investigation, this study offers mostly simple distribution data for camel crickets. These data may also be compromised by the collection methods (unconfirmed citizen scientist records and uneven distribution of survey responses). There is nothing here about the ecology of camel crickets in urban areas - just records of insects in homes. I believe that the paper could be improved by additional information and investigations.

·

Basic reporting

The project described in this paper is a good example of a situation in which citizen science can provide data not easily collected by other methods, namely biodiversity in private homes. The camel cricket project described here, as well as the broader program in which it is contained, have the potential to contribute solid data to studies of urban biodiversity. The focal organisms in this case are some of the few arthropods found in homes that are both easy to spot, and easy to identify to species based on good photographs. Those qualities make them well-suited to citizen science monitoring. This paper is likely to be as useful as a test case in how to do citizen science with arthropods, as it is in understanding changing camel cricket distributions and populations. The study also contributes useful natural history data on these commonly encountered species.

The overview of the habitats and distribution of the camel crickets is generally good, but I think some expansion on diet, life history, behavior and interactions with other arthropods likely to be found in houses might be helpful, partly to place the results in context, and partly for readers who are interested in the broader context of arthropod diversity in homes.

The data in Table 1 might be better visualized in map form. Maps are used to good effect for other data, and having, for example, the response rate and percentage of houses with crickets superimposed on a map of the states themselves would allow readers to more easily see where records were concentrated. With a little thought I’m sure a visually appealing map that incorporates the numerical data could replace the table.

In Table 2, I think the last three columns might be more intuitive if they were broken down into D. asynamora / D. japonica / unidentified Diestrammena

There are a few formatting problems in the References (format of authors’ names, italicization of scientific names, etc.) These should be checked carefully. Vickery & Kevan (1967) is cited in the text but not listed in the references.

Experimental design

The research question is simple but relevant, given the growing interest in urban ecology and urban biodiversity. The design of the study is reasonably well-described. The methods are mostly appropriate and the results and conclusions usually do not overstep the data (but see comments on the pitfall trap component).

Validity of the findings

Any study based on voluntary survey data is going to be subject to bias and unequal sample size, but that’s just the nature of the game in citizen science. The authors have done a good job of identifying the weaknesses and staying within the limits imposed by the uneven response. Some of the analyses and interpretations become more questionable as distance from North Carolina increases, and I wonder if some of the quantitative results might be more rigorous if they were restricted to eastern or southeastern states where the response was higher.

I’m not convinced that the pitfall trap transect survey data from the yards is rigorous enough to say much about patterns within yards or between yards and houses. There was a single trap per distance per yard, and the total number of yards was small, especially when negative results are factored in. In addition, the spacing between traps (especially 1m and 4m) is likely too small to be biologically meaningful (just a few cricket leaps apart). There is no description of the yards, so there is no way of knowing if, for example, the far traps tended to be closer to a fence or gardens, or how different the yards were (lawn versus gardens versus mulch versus gravel, sun versus shade, etc.). I like the inclusion of the trap component in saying something about whether the crickets are outside as well as inside the houses, but I would be very cautious about reading too much into the pattern along a linear transect. Similarly, collecting the crickets in pitfall traps outside versus spotting them inside will not tell you anything about whether they are living inside/foraging outside or vice-versa. To get at this, you would need to observe feeding behavior, as well as being able to determine (using marking) whether you were tracking the same crickets inside and outside.

---

## Round 0.2 · accepted · Accept

Thank you for your rebuttal and revisions. In my opinion, the MS is now acceptable for publication in PeerJ.

One small comment – the photograph of the D. japanica individual in Figure 4b is fairly blurry. It is still possible to make out the overall difference between the two species, but I'd say that this image is on the far edge of the acceptable-blurriness spectrum. If you have a better image (or perhaps a higher resolution version of this image?), it would be great if you could use it instead.

I would like to encourage you to make the peer review history of this MS public as well.

Thanks again.